# Structure of an RNA G-quadruplex from the West Nile virus genome

J. Ross Terrell[1,3], Thao T. Le [1,3], Ananya Paul[1], Margo A. Brinton [2], W. David Wilson[1], Gregory M. K. Poon [1], Markus W. Germann [1,2] & Jessica L. Siemer [1]

Potential G-quadruplex sites have been identified in the genomes of DNA and RNA viruses and proposed as regulatory elements. The genus *Orthoflavivirus* contains arthropod-transmitted, positive-sense, single-stranded RNA viruses that cause significant human disease globally. Computational studies have identified multiple potential G-quadruplex sites that are conserved across members of this genus. Subsequent biophysical studies established that some G-quadruplexes predicted in Zika and tickborne encephalitis virus genomes can form and known quadruplex binders reduced viral yields from cells infected with these viruses. The susceptibility of RNA to degradation and the variability of loop regions have made structure determination challenging. Despite these difficulties, we report a high-resolution structure of the NS5-B quadruplex from the West Nile virus genome. Analysis reveals two stacked tetrads that are further stabilized by a stacked triad and transient noncanonical base pairing. This structure expands the landscape of solved RNA quadruplex structures and demonstrates the diversity and complexity of biological quadruplexes. We anticipate that the availability of this structure will assist in solving further viral RNA quadruplexes and provides a model for a conserved antiviral target in *Orthoflavivirus* genomes.

The genus *Orthoflavivirus* within the family *Flaviviridae* contains arthropod-transmitted, positive sense, single-stranded RNA viruses. Mosquito-borne orthoflaviviruses including dengue virus (DENV), Japanese encephalitis virus, West Nile virus (WNV), yellow fever virus and Zika virus (ZIKV), cause over 400 million reported human infections globally[1]. In addition, newly emerging viruses such as Usutu virus and the tick-borne Powassan virus (POWV) are becoming more prevalent in Europe and North America, respectively. Despite causing significant global disease, anti-orthoflavivirus treatments remain limited, with few vaccines available to humans and no antivirals currently on the market. Popular strategies for developing *orthoflavivirus* antivirals have targeted the enzymatic functions of viral proteins, such as the polymerase (NS5) or protease (NS3) or have attempted to disrupt host-virus protein interactions[2,3]. The focus on protein interactions has

been in part due to the availability of structural information for viral proteins garnered through NMR[4], x-ray crystallography[5], and cryo-EM[6].

However, the study of RNA structures in viral genomes[7-11], has recently expanded revealing their surprising complexity and potential as antiviral targets. Nucleic acid interactions have previously been shown to regulate *orthoflavivirus* lifecycles. For example, long-range RNA base pairing facilitates the circularization of the genome and subsequent synthesis of the negative-sense template strand[7]. Stem-loop structures, particularly in the 3′ and 5′ regions, have also been proposed to recruit both host[7] and viral proteins[8,9]. Nucleic acid sequences can also adopt complex structures via association of canonical and non-canonical pairing of nucleobases. For example, pseudoknots have been proposed as structural features with important functions during the orthoflavivirus lifecycle[10,12,13]; while higher-

[1]Department of Chemistry, Georgia State University, Atlanta, GA 30303, USA. [2]Department of Biology, Georgia State University, Atlanta, GA 30303, USA. [3]These authors contributed equally: J. Ross Terrell, Thao T. Le. ✉e-mail: mwg@gsu.edu; jsiemer1@gsu.edu

order RNA associations may contribute to genome compaction and viral packaging[8].

G-quadruplexes are another category of complex nucleic acid structures that may be significant in the regulation of the *ortho-flavivirus* lifecycle. G-quadruplexes form in G-rich DNA or RNA when four guanines form hydrogen bonds to create Hoogsteen geometry and form a tetrad. These tetrad structures are further stabilized by a monovalent cation, usually potassium, and π-stacks forming a stable quadruplex. Since the first detailed reports of quadruplex structures in *Tetrahymena* telomeric DNA[14], followed by one in *E. coli* 5S RNA[15], numerous studies have demonstrated the presence of G-quadruplexes in a diverse range of organisms and their significant roles in the regulation of cellular processes, such as transcription and genome stability[16–18]. Computational studies have identified potential G-quadruplex sites in many viral genomes[18–21] including those of the genus *Orthoflavivirus*, with multiple conserved G-quadruplex sites predicted across the coding regions of these genomes[22,23]. Further biophysical studies have shown that some of the predicted G-quadruplex sequences in the ZIKV, tickborne encephalitis virus (TBEV) and WNV genomes are able to form quadruplexes in vitro[22,24–28], and treatment of ZIKV or TBEV infected cells with known quadruplex binders has an antiviral effect[25–27].

In this work, we report the crystal structure of a viral genomic RNA G-quadruplex, which was derived from the sequence in the NS5 gene of WNV. There is current interest in exploring viral quadruplexes as potential antiviral targets; however, structural information on RNA quadruplexes required for rational drug design is very limited[20,29,30]. Additionally, the functional roles of *orthoflavivirus* G-quadruplexes are currently unknown; however, they may be involved in regulating transcription and/or translation, improving genome stability or facilitating genome compaction and packaging. Analysis of this crystal structure reveals a two-stack G-quadruplex (α/β tetrads) that is stabilized by two capping structures (γ-triad and A6 • A20 dyad) and noncanonical base pairing with loop residues. The unusual features of the WNV NS5-B quadruplex expand our knowledge of quadruplex complexity and will aid in designing molecules to target them.

## Results

### A predicted Viral RNA quadruplex forms in vitro

We experimentally validated the formation of the predicted WNV RNA quadruplex NS5-B, first with a 30- and then 27-mer sequence, using gel electrophoresis and staining with the quadruplex specific dye, Thioflavin T (ThT). We confirmed monomer formation using mass spectrometry (Fig. S1a). To proceed with structure determination, the sequence was further reduced to a 21-mer (Fig. 1a) and characterized in solution using a series of established biophysical methods. NMR proton spectra of NS5-B (Fig. S1b) showed characteristic imino proton signals at 10–12 ppm from the G-tetrad Hoogsteen base pairing. Eight proton signals arising from the imino protons of two stacked G tetrads are expected in this region. However, more peaks of different intensity were observed indicating the presence of multiple conformations. To improve NMR peak resolution further, we designed a modified RNA sequence, NS5-B M1 with G→A substitutions near the 5′ and 3′ termini (Fig. 1a). NS5-B M1 exhibited the same stability and a similar but better-resolved imino proton spectrum, indicating less conformational variability of the monomeric structures (Figs. 1d, S1b). Both NS5-B wild-type (WT) and M1 fold into compact structures that exhibit similar electrophoretic mobility on native PAGE and stain equally with ThT (Fig. S1a). Mass spectrometry (Fig. 1b) confirmed the presence of a single ~7 kDa species, corresponding to a monomeric 21-mer in agreement with the electrophoretic mobility. We further confirmed the formation of a monomeric species for M1 and WT through NMR concentration dependence studies (Fig. S2). The CD spectra of both RNAs are essentially identical and are characteristic of a parallel G-quadruplex topology with $I_{max}$ at ~264 nm and $I_{min}$ at ~245 nm

(Figs. 1c, S1c). Therefore, given the highly similar behavior of NS5-B WT and NS5-B M1, either quadruplex was considered relevant for further studies. However, neither sequence yielded an NMR spectrum that was resolved enough for structure determination, but NS5-B M1 formed suitable crystals that enabled us to solve the x-ray crystal structure of a WNV genomic RNA quadruplex (Fig. 1e, Table 1 and Movie S1).

### High resolution X-ray structure of an *Orthoflavivirus* genomic G-quadruplex

The x-ray crystal structure of NS5-B M1 was resolved to 1.97 Å and contains two non-equivalent quadruplex copies per asymmetric unit, Chains A and B, packed head-to-head (Fig. S3). Each structure forms a monomer with no base-dependent contacts across the planar interface (Fig. S3). Each monomer contains two G tetrads arranged in a right-handed parallel strand configuration. In both structures, the first tetrad (α) is composed of nucleotides G4-G10-G13-G17, and the second tetrad (β) is formed by G5-G11-G14-G18 (Fig. 2a). A third set of stacked bases forms a triad (γ) composed of G8-U15-U19 and is further capped by the A6 • A20 dyad (Fig. 3a). The dyad and γ-triad are connected by A7, A9 and U16 connect the γ- triad to the α-tetrad, and an additional loop connects the α-tetrad to the β-tetrad by C12 (Fig. 2b). The G tetrads have both north and south sugar pucker conformations, and the bases adopt an *anti*-conformation (Table S2). A $K^+$ ion is coordinated between the α- and β-tetrads in both chains, and $NH_4^+$ ions are present between the β-tetrad and γ-triad in both Chain A & Chain B (Fig. 1e) as well as at the interface between the copies (Fig. S4).

To further investigate the novel structure of NS5-B M1, we performed two, 1-μs molecular dynamics (MD) simulations: one with NS5-B M1 and one with the wildtype sequence. We initialized both simulations using the x-ray structure coordinates of NS5-B M1 and ran the simulations in explicit water with 0.15 M KCl present. In both the M1 and WT G4 structures, the α- and β- tetrads remain highly stable (Figs. S5 and S6). Specifically, the Gs form long-lasting H-bonding with a bond distance of approximately 3.0 Å (Figs. S5a and S6). This confirms that the tetrads remain stable and intact, independently of the mutated regions in the 3′ and 5′ terminal sequences.

### Unusual terminal capping structures facilitate two-stack quadruplex formation

Tetrad π -stacking is a main driver of quadruplex stability. Therefore, two-stack quadruplex formation would be predicted to be less energetically favorable than stacks of three or more tetrads. However, while most of the deposited quadruplex crystal structures are composed of three or more stacked tetrads, several two-stack tetrad structures have been observed[30–33], with many possessing features such as capping structures and interactions with loop bases to enhance stability. Some examples of two-stack RNA tetrads with additional stabilizing features are Mango-III[34] and Spinach[35,36] RNA aptamers. In the Mango-III aptamer, two G-tetrads stack on a noncanonical terminal U-A-U base triple and are further capped by a U-A base pair at the top of the tetrads[34]. Similarly, the Spinach aptamer has a two-stack G-tetrad[35,36] which stacks on a noncanonical G-U-U-C tetrad[36] and is further capped by a U-A-U base triple. Although the monomeric NS5-B M1 is structurally distinct from the vegetable family of aptamers that form in duplex RNA, its stability is also markedly enhanced by unusual terminal structures.

One stabilizing feature is the γ-triad, a stacked planar capping structure, (Fig. 3a), which arises from a unique extension of A6 in the parallel A7-loop followed by a foldback of G8 (Fig. 3b). G8, U15, and U19 form noncanonical bifurcated wobble hydrogen bonding[32] between G8 and U19 as well as a single hydrogen bond between U19 and U15[33] (Fig. 3a). To further examine the stability of the G • U and U • U bonds, we observed the 1-μs MD simulations for both the M1 and WT sequences. We observe long-lasting H-bond stability for both sequences (Fig. S7) suggesting that triad stability is also independent

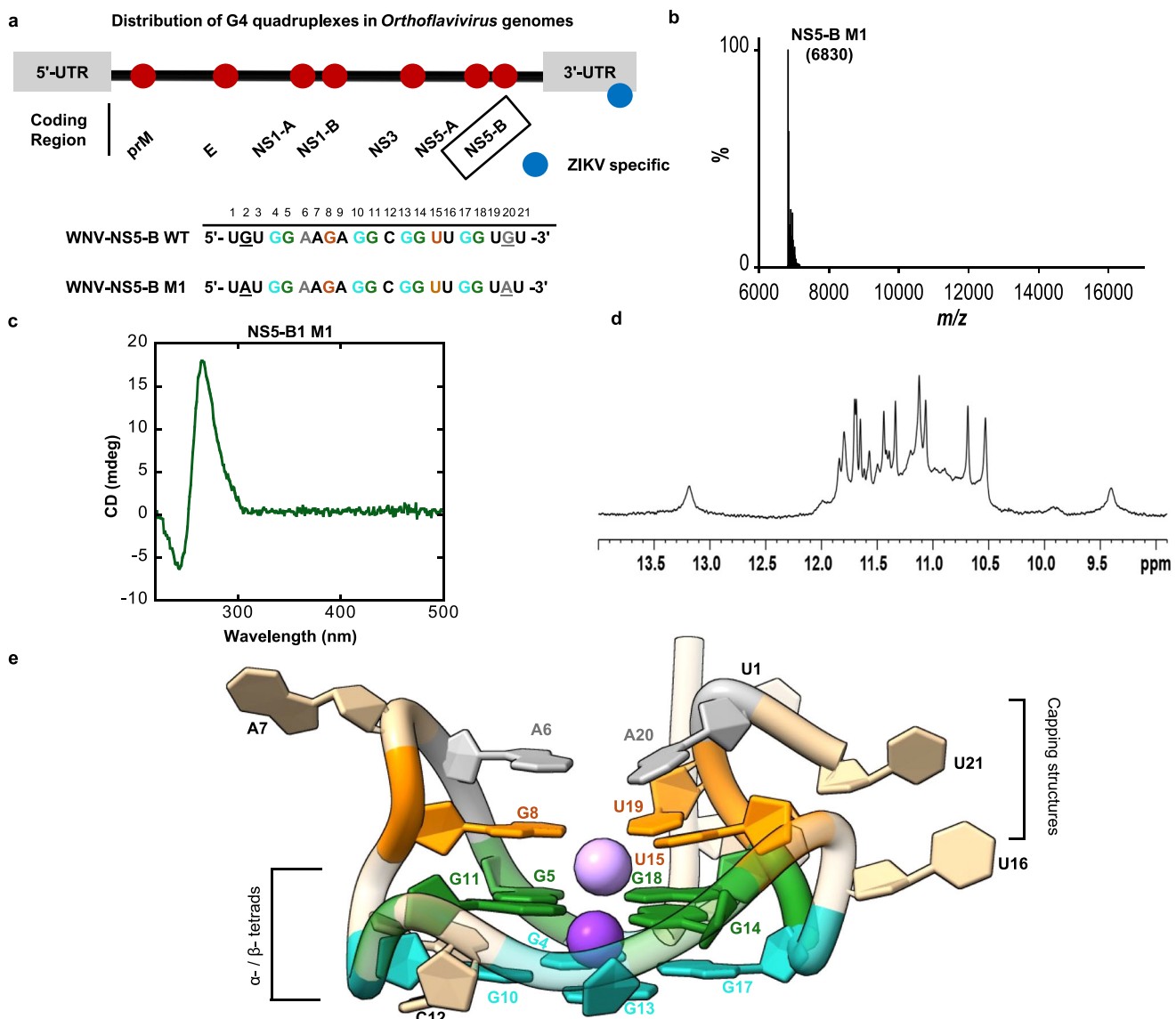

**Fig. 1 | Solution state characterization of NS5-B M1. a** Location and sequence of the NS5-B in an *Orthoflavivirus* genome (top). Wild-type (WT) and mutated (M1) NS5-B sequences (bottom). Bases making up the α-tetrad are in teal, the β-tetrad are in green, the γ-triad are in orange, and the dyad in gray. Sequence variation is indicated by underlining. **b** Mass spectrometry confirms that NS5-B M1 is a monomer with a mass of ~7 kDa. **c** The CD spectrum is characteristic of parallel topology with a maximum at ~264 nm and a minimum at ~245 nm. **d** The 1D $^1$H imino proton spectrum of NS5-B M1 at 298 K in the presence of K$^+$ ions exhibits characteristic quadruplex signals at 10–12 ppm and is consistent with the spectrum of the WT sequence (Fig. S1a). **e** A view of the 1.97 Å x-ray crystal structure of the G2A/G20A mutated NS5-B M1 West Nile Virus quadruplex. Additional views are provided in Movie S1. Source data for (**b**, **c**) are provided as a Source Data file.

of the 3′ and 5′ terminal sequences. A second terminal capping structure, the A6 • A20 dyad (Fig. 3a), occurs on top of the γ-triad. While this interaction involves a mutated residue, the NMR, CD thermal melting and MD data demonstrated that these mutations cause little effect on the global structure. The MD simulations also demonstrated that an unusual base pair can form at the 3′-side of A20 • A6 in M1 and G20 • A6 in WT. These base pairs are dynamic but are intermittently stabilized by non-canonical hydrogen bonding (Fig. S8). Interestingly, a dyad-triad capping structure also exists in the two-stack DNA quadruplex from the human telomere sequence[32]. The dyad-triad capping structure in NS5-B M1 similarly provides more stability through additional π-stacking and is also able to facilitate crystal packing through additional inter-asymmetric unit hydrogen bonding between A20 and A2 (Fig. S2).

To further examine the role of these capping structures in maintaining quadruplex stability, we designed and biophysically characterized three triad and dyad mutants (Table S3). The mutants with 3′

terminal deletions, M1 D2 and M1 D3, and the substitution mutant, M1 G8A, all formed monomeric quadruplex structures with different degrees of compaction, as indicated by their varying mobilities on Native PAGE (Fig. S9). NMR imino proton spectra of the mutants exhibited characteristic G-tetrad signals at 10–12 ppm (Fig. S10). Noticeably, while the M1 D2 spectrum retains a signal around 12.8 ppm, there were no signals observed in the Watson-Crick base pairing region of the spectra for either M1 D3 or M1 G8A. These data suggest that the imino proton signals in the 12–14 ppm region likely arise from the G • U base pair in the γ-triad. Since the biophysical characterization demonstrated that all mutants were able to form quadruplexes, we conclude that the two-stack tetrad in NS5-B M1 is sufficiently stable to form without the terminal capping structures (Fig. S11). CD thermal denaturing analysis was performed to quantify the contribution of each capping structure to the overall structural stability. CD thermal melting revealed minimal differences between the M1 and WT stability ($T_m$ 48 ± 1 °C and $T_m$ 47 ± 1 °C), and the thermal melting of M1 D2 ($T_m$ 46 °C ± 1 °C) showed

**Table 1 | Crystallographic and refinement statistics (PDB ID: 8UTG)**

| | NS5-B M1 – G-Quadruplex |
|---|---|
| **Data collection[a]** | |
| Space group | P3₂ 2 1 |
| Cell dimensions | |
| $a$, $b$, $c$ (Å) | 33.8049, 33.8049, 163.376 |
| a, b, g (°) | 90, 90, 120 |
| Resolution (Å) | 29.28–1.97 (2.04–1.97)[b] |
| $R_{merge}$ | 0.139 (3.254) |
| $I$ / $sI$ | 11.7 (2.8) |
| Completeness (%) | 99.9 (100) |
| Redundancy | 14.1 (14.8) |
| **Refinement** | |
| Resolution (Å) | 29.28–1.97 |
| No. reflections | 8317 (790) |
| $R_{work}$/$R_{free}$ | 0.1937/0.2426 |
| No. atoms | |
| RNA | 908 |
| Ligand/ion | 17 |
| Water | 22 |
| $B$-factors (Å²) | 47.08 |
| RNA | |
| Ligand/ion | 24.98 |
| Water | 44.90 |
| R.m.s. deviations | |
| Bond lengths (Å) | 0.015 |
| Bond angles (°) | 1.88 |

[a] Diffraction data was collected from a single crystal.
[b] Values in parentheses are for the highest-resolution shell.

only a minor decrease. These data confirm that changes to or disruption of the dyad have minimal effect on the quadruplex stability. However, the thermal melting of M1 D3 ($T_m$, 45 °C ± 1 °C) and M1 G8A ($T_m$, 41 °C ± 1 °C) showed significant reduction in melting temperature compared to M1 and WT, indicating that the triad has a major contribution to the stability of this quadruplex structure.

**Loop dynamics further stabilize tetrad formation via transient interactions**

It was previously thought that to resolve steric clashes with the 2′ OH, RNA G-quadruplexes were governed by similar stereochemical rules as A-form duplex RNA, resulting in a bias towards parallel topology, G tetrads with an *anti*-conformation and a preference towards *C3′ endo* sugar puckering[37]. However, the structures for several RNA aptamers have shown that RNA quadruplexes have much wider conformational freedom and use many different structural features to maintain stability[30,33–35]. Consistent with this, we observe several interesting stabilizing features in the loops of NS5-B M1.

A key feature of the G-quadruplex architecture in the NS5-B M1 structure is the length of the loops and segments between capping structures. The C12 loop consists of a single, one nucleotide segment (Fig. 3c). The A6-A7-G8-A9 loop and the U15-U16 loop contain single nucleotide segments (U15, A7 and A9) between the capping structure residues. We hypothesize that the short nucleotide segments contribute favorably to the overall energetics[31] and the crystal packing of this structure by constraining the placement of nucleotides in the triad and tetrad regions thereby facilitating optimal base stacking. To maintain the parallel topology in the NS5-B M1 structure, the A6 and A7 bases located at the top of the quadruplex create a striking structural feature. While A6 is involved in the terminal dyad capping structure, A7

makes a short turn to allow G8 to occupy a more favorable stacked position in the γ-triad. This generates a 4-purine stack G4-G5-G8-A6 (Fig. 3b) which we hypothesize also forms in the WT structure given the similar melting temperatures of the M1 and WT quadruplexes and its persistence in an MD simulation of the WT sequence.

Another surprising structural observation is that two of the loop residues, A9 and C12, interact directly with the α-tetrad. A9 forms a double-chain-reversal loop that connects the α-tetrad and the γ-triad. From this position, A9 forms a pentad with the α-tetrad, A9 • (G4 • G10 • G13 • G17), via an A9 • G4 N7-amino hydrogen bond in Chain A, and an additional A9 • G4 amino-N3 hydrogen bond[38] in Chain B (Fig. 2b). In addition, the positioning of A9 at the groove releases water molecules. In the initial MD simulation of M1, A9 remained perpendicular to G4, without any significant close contact for approximately 400 ns (Fig. S5c). However, after equilibration, during the remaining simulation time (600 ns), A9 forms two stable H-bonds with G4, with a bonding distance of approximately 3.0 Å. This result is consistent with the x-ray crystallographic configuration. For the WT simulation, A9 also frequently formed two H-bonds. However, the H-bonds for WT (A9-N7 with N2-G4) are more dynamic (Fig. S5c). This conformational flexibility is consistent with the broad signal observed in the WT NMR spectra (Fig. S1a).

In contrast with the stability of A9, both the crystal structure and MD simulations reveal that C12 has a wider conformational variability. Two distinct C12 conformations were observed in Chains A and B of the structure (Fig. 2b). In the "flipped out" position (Chain B), the base extends into solvent and exhibits high B-factors (Fig. 3c). In the "flipped in" position (Chain A), C12 forms two nonconical C12 • G10, N3-amino, amino-N3 hydrogen bonds[39]. The positioning of the C12 residue affects the global structure of the quadruplex resulting in G10 alternating sugar puckering conformations between Chains A and B of the crystal structure (Table S2) as well as during MD simulations. In the M1 simulation (Fig. S5b), the frequency of the "flipped in" position is relatively high. The simulation of the WT sequence also shows a high frequency of H-bonding interactions but is also more dynamic (Fig. S5b) compared to M1 which may have facilitated crystallization of the M1 quadruplex.

## Discussion

**Three levels of pressure maintain NS5-B M1 conservation in *Orthoflavivirus* genomes**

High conservation of multiple potential G-quadruplex sites across the genome coding region appears to be a unique feature of *orthoflaviviruses*. In contrast, most other viruses have potential quadruplex sites in the 5′ and 3′ UTRs of their genomes[40]. The presence of quadruplex sites in the genomic coding region is particularly advantageous for antiviral targeting due to multiple levels of conservation pressure. The NS5-B quadruplex sequence remains highly conserved with minor changes in loop residues across a diverse set of both mosquito-borne and tick-borne *Orthoflavivirus* members (Fig. 4a, b). Further analysis suggested that three levels of conservation pressure apply to this sequence region: (1) the nucleic acid structure, (2) the protein structure, and (3) the protein function. While the function of quadruplexes in *Orthoflavivirus* genomes is currently unknown, it seems likely that they have a regulatory role in the orthoflavivirus lifecycle given their preservation among members of the genus which suggests importance for viral fitness. The sequence of NS5-B also has an impact at the protein-coding level as it is in the region that encodes the methyltransferase domain of the viral NS5 polymerase. At the amino acid level, this region is highly conserved with the only variation occurring at position 7 (serine or cysteine) (Fig. 4b). This conservation is likely due in part to the maintenance of the alpha helix and flexible loop secondary structure of the protein. However, this part of the protein structure is also enzymatically relevant as it is a putative SAH-binding site[5]. Therefore, both the structure and function of the methyltransferase limit variation in the amino acid sequence and

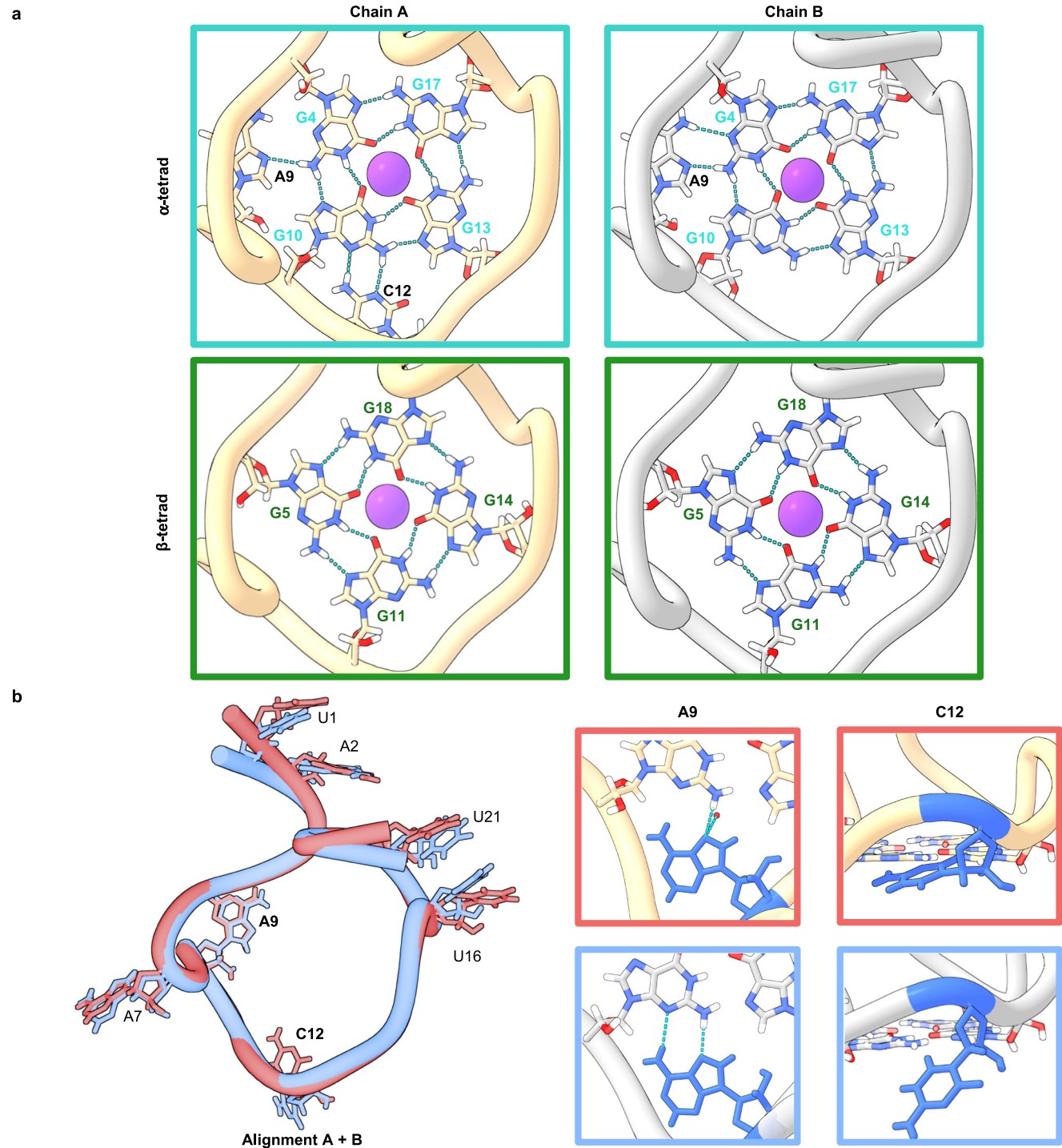

**Fig. 2 | The configuration of the two-stack G-quadruplex provides an unusual topology with triplet formation stabilizing the bases directly above the top G tetrad. a** Guanosine interactions in the α- and β-tetrads and the loop-base to tetrad stabilization of the α-tetrad. Chain A (left) shows the α-tetrad is stabilized through noncanonical hydrogen bonding in the A9 loop between A9 and G4 and the C-loop between C12 and G10. In Chain B (right) the α-tetrad is stabilized through noncanonical hydrogen bonding in the A9 loop between A9 and G4. **b** Alignment of

Chain A in red and Chain B in blue (left) illustrates overall structural similarities between the two copies. Residues with a high degree of overlap have been omitted. A9 and C12 have minor changes in bonding interactions (right). In Chain A, C12 (blue) is observed in the downward position contacting the G10 of the α-tetrad while in Chain B, C12 (blue) is observed in the upward position extending into the solvent environment.

consequently the nucleic acid sequence, which also preserves the quadruplex. Targeting of features that have multiple levels of conservation pressure such as the NS5-B quadruplex is a preferential antiviral strategy as it limits the ability of the virus to escape through mutation.

## Expansion of the RNA quadruplex structural landscape is essential for therapeutic design

Despite the rapid expansion in understanding the importance and versatility of RNA quadruplexes, there are still limitations in predicting the presence and structures of quadruplexes in sequences of

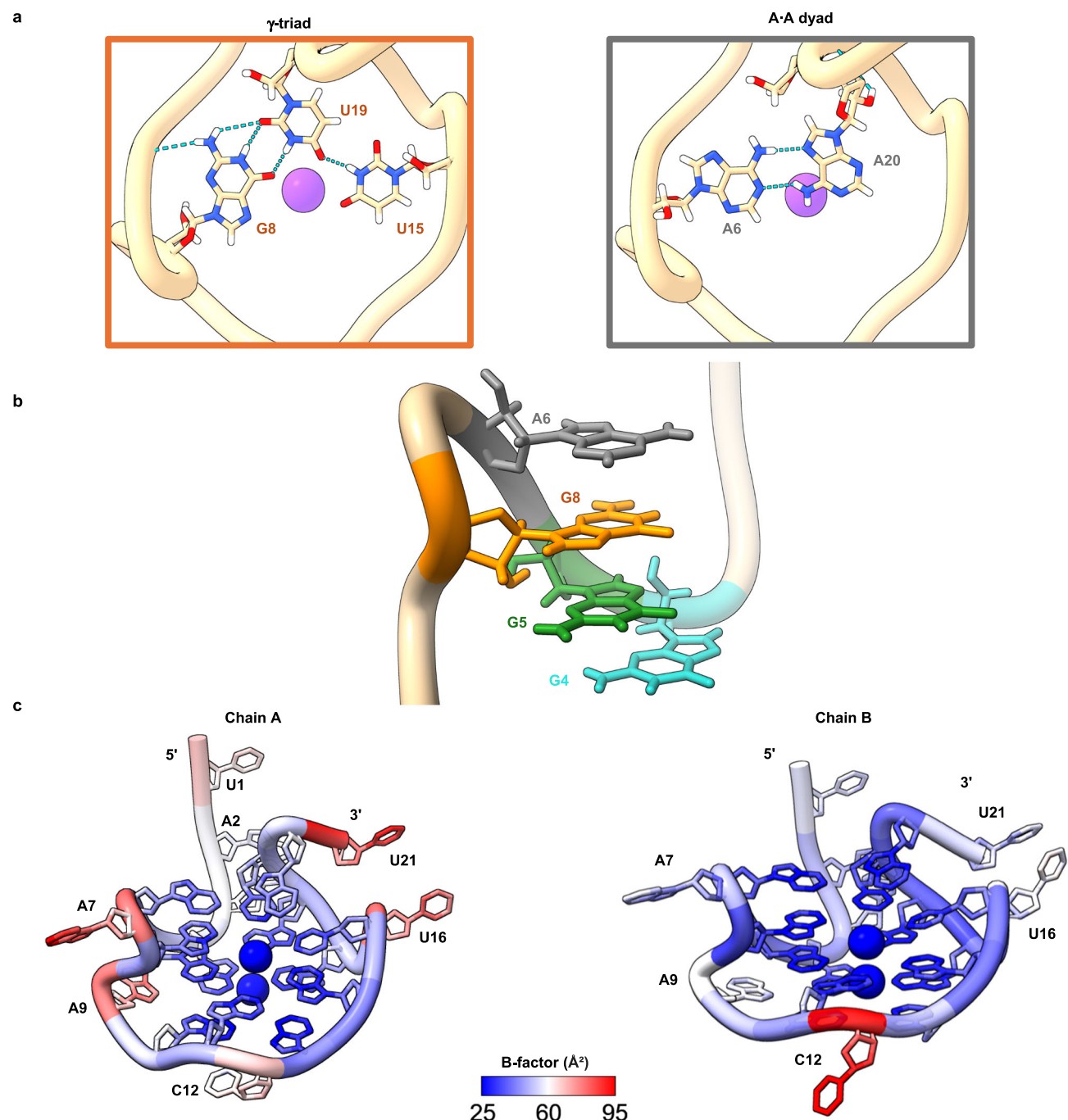

**Fig. 3 | Loop structures show a range of dynamics. a** Analysis of the γ-triad reveals unusual G•U•U base pairing interactions: a bifurcated wobble hydrogen bonding G8•U19 O6-imino, imino-O2 and a U19•U15 O4-imino hydrogen bonding (left). The A6-A7-G8-A9 loop forms a foldback, which facilitates the formation of noncanonical A•A base pairs on the top of the quadruplex, denoted as the A'A dyad (right). **b** Four vertically stacked purine bases with A6 in gray, G8 in orange, G5 in green and G4 in teal. **c** Three of the quadruplex loops facilitate prominent structural features: the γ-triad, a noncanonical stacked base pair, and a dynamic stabilization of the α-tetrad through hydrogen bonding.

interest[17,41]. Experimental methods for probing quadruplexes in the context of longer biological sequences are limited by imaging capabilities and the specificity of quadruplex-specific antibodies and fluorescent stains and do not provide structural details. The small number of available high-resolution RNA quadruplex structures limit the development of computational tools, and existing methods are unable to predict less common structural features such as those that are present in the NS5-B M1 structure. The availability of experimentally determined structures provides essential parameters for developing improved quadruplex prediction algorithms.

We used ESI-MS to observe the binding of two commercially available end stackers, BRACO-19 and PDS, previously reported to stabilize ZIKV G4 forming sequences in vitro and inhibit viral replication in ZIKV-infected cells[25,26]. Both small molecules bind NS5-B M1 at a 2:1 ratio despite the non-equivalent interfaces at the top and bottom of the quadruplex (Fig. S12). While these data support end stacking[42] as a viable approach for targeting RNA quadruplexes, improved binding specificity to unique features would greatly increase the ability of quadruplex binders to be used as therapeutics[43]. We hypothesize that the unusual structural features of NS5-B M1 reported in this study

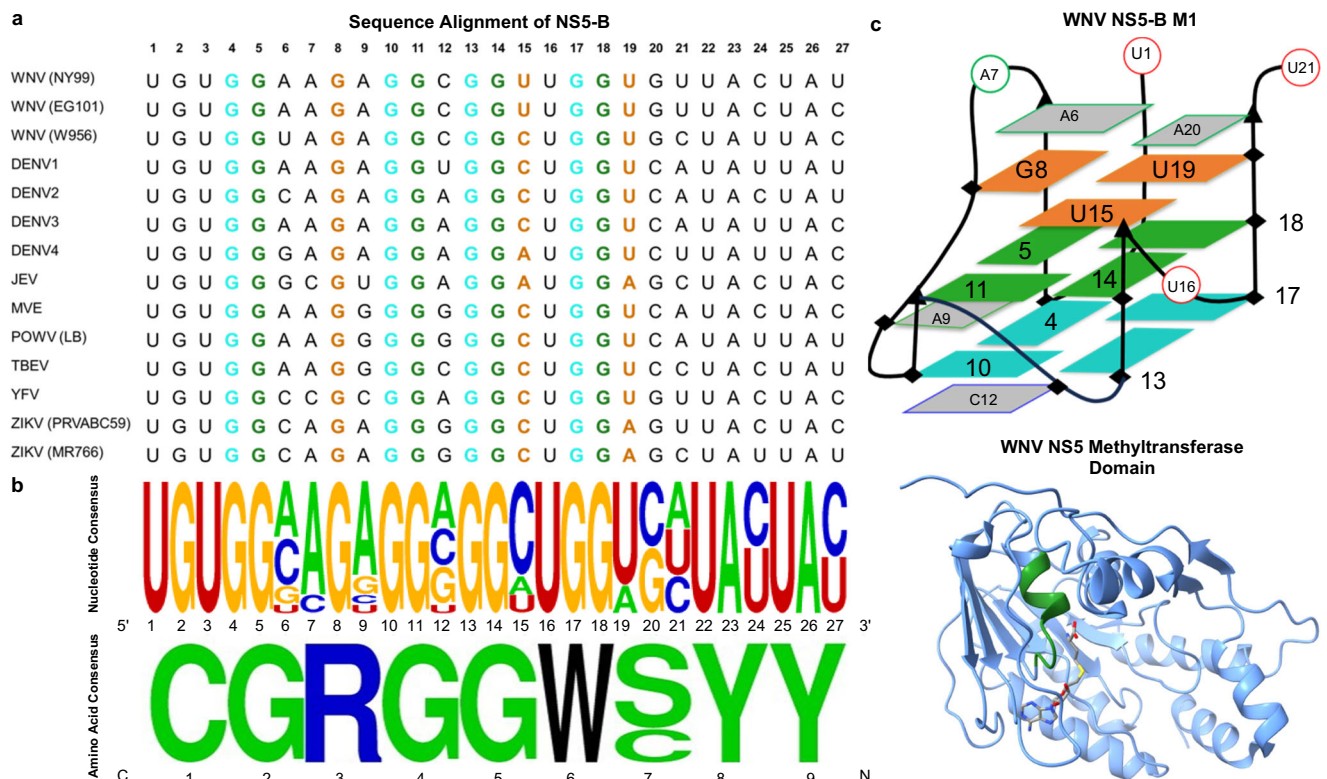

**Fig. 4 | Conservation of the NS5-B quadruplex sequence. a** The nucleic acid sequence alignment for the NS5-B region in the genomes of members of the genus *Orthoflavivirus* (Table S4). **b** The nucleic acid consensus sequence (top) and the amino acid consensus sequence (bottom) show high conservation across phylogenetically diverse *Orthoflavivirus* members (Table S4). This conserved region highlighted in green on the NS5 methyltransferase structure (PDB: 2OY0) encodes a putative SAH-binding site[5] (right). **c** Schematic representation of the WNV NS5-B M1 quadruplex structure (PDB: 8UTG).

present an opportunity for increased specificity of targeting this (and similar) quadruplex structures. The two capping structures, the dyad and triad, represent interfaces (Fig. 4c) for the development of anti-virals that are potentially both specific and panflaviviral. The use of peptide nucleic acid specifically designed to target WNV quadruplexes at femtomolar affinity demonstrates the power of specific sequence targeting[28]. However, production and delivery of these types of therapeutics are challenging. Therefore, further screening of libraries of RNA binding small molecules using high-throughput methods such as microarrays[44,45], surface plasmon resonance[46], fluorescent intercalator displacement assay[47], and in silico simulations[48–50] continue to be essential strategies for developing the next generation of RNA virus therapeutics.

In summary, the resolution of this orthoflaviviral quadruplex structure is an important step in advancing the study of additional RNA quadruplexes relevant to human disease. The wealth of novel features found in this structure provide input for the refinement of current in silico methods, a basis for solving similar quadruplex crystal structures, and information for the rational design of novel therapeutics.

## Methods

### Oligonucleotides
RNA sequences (Supplementary Table 1) without a 5′ triphosphate were obtained from Millipore Sigma. The RNA was resuspended in RNase free diH2O. For each experiment, stock RNA was diluted into the respective buffer to the appropriate concentration and heated to 95 °C for 5 min. The samples were cooled and subsequently stored at 4 °C overnight.

### Gel Electrophoresis
Sequences were visualized on a native 15% 19:1 polyacrylamide gel made and run using 1X TBE buffer containing 10 mM KCl.

Bromophenol blue was used as the dye marker. Images were captured using UV shadowing and by staining with 10 μM Thioflavin T (Sigma Aldrich).

### CD
NS5-B M1 was prepared at 4 μM concentration in 20 mM potassium phosphate, 50 mM KCl, and 0.5 mM EDTA at pH 6.5. CD spectra were recorded at 25 °C from 500 to 220 nm on a JASCO 1500 CD spectrometer using a scan speed of 50 nm/min and a response time of 1 s. For each sample, four spectra were averaged. Buffer-subtracted graphs were created using the Kaleidagraph 5.0.6 software.

For CD melting experiments, spectra were collected on as JASCO 1500 CD spectrometer for WNV NS5-B WT and M1 as well as the M1 D2, D3 and I3 mutants (Table S3). Samples were prepared at 4 μM in 20 mM potassium phosphate, 50 mM KCl, and 0.5 mM EDTA at pH 6.5. Melting curves were obtained by heating the samples from 20 to 90 °C at a rate of 1 °C/min using integration time of 4 s. The melting process was monitored by recording the CD spectra at wavelengths ranging from 220 to 400 nm. To generate a CD melting curves, a single wavelength of 264 nm corresponding to the parallel G-quadruplex was monitored and normalized using the following equation: (Abst − min)/(max − min), where Abst is the absorbance at a given temperature, max is the maximum absorbance at 264 nm, and min is the minimum value. Spectra and the thermal melting plot were created using Graphpad 10.1.2.

### NMR
[1]H NMR spectrum was collected on a Bruker AVI 500 MHz NMR equipped with a 5 mM TBI probe. RNA samples were diluted with NMR sample buffer containing 20 mM potassium phosphate, 50 mM KCl, 0.5 mM EDTA, and 10% D2O to achieve a final concentration between

100–200 μM. Samples were adjusted to pH 6.5. 1D $^1$H imino spectra were acquired using 1–1 jump-return solvent suppression.

For the concentration dependence $^1$H NMR analyses of NS5-B WT and M1 spectra were collected on a Bruker Avance 600 MHz NMR equipped with a 5 mm QXI probe. $^1$H imino spectra were acquired using a modified jump and return pulse sequence at 297 K. RNA samples were diluted with NMR sample buffer as described above. Sample concentrations ranged from 500 μM to 12 μM. $^1$H spectra of the aromatic region were acquired using presat pulse sequence at 297 K. RNA samples were diluted with NMR sample buffer in 100% D$_2$O, pH* 6.4 and ranged from 130 μM to 5 μM concentrations for WT or to 1 μM for M1 respectively.

TopSpin 4.1.4 software was used for data processing.

### Electrospray Ionization Mass Spectrometry (ESI-MS)

NS5-B M1 was dialyzed in 100 mM NH$_4$OAc and samples were prepared at concentrations ranging from one (10 μM) to three (30 μM) equivalent of BRACO-19 or Pyridostatin (Sigma Aldrich) to NS5-B M1 RNA supplied with 5% MeOH. ESI-MS analyses were performed on a Waters Xevo G2_XS Mass Spectrometer (Waters Corporate, Milford, MA) equipped with an electrospray ionization source in positive ion mode with the capillary voltage of −2000 V and cone voltage of −30 V with desolvation at 350 °C and source temperature at 120 °C. NS5-B M1 was injected using an autosampler at 50 μL/min flow rate and ESI-MS spectrum was obtained using full scan analyses. Data acquisition and processing were performed using MassLynx 4.2 and Kaleidagraph 5.0.6 software.

### Crystallization

A solution of annealed G-quadruplex was prepared at 1 mM concentration in 50 mM KCl, 20 mM potassium cacodylate pH 6.5, 200 mM beta-mercaptoethanol buffer. The stock solution was screened at a 1:1 ratio against the Helix HT 96-well crystallization screen[51] (Molecular Dimensions) using an Art Robbins Crystal Gryphon robot in trays with a 3-well sitting-drop configuration. Crystal growth was observed approximately 4 weeks after trays were prepared with condition F4 yielding small but sharp ~15 × 15 × 25 micron trigonal bipyramidal crystals. Crystals grown in F4 wells (50 mM KCl, 50 mM Bis-Tris pH = 7.0, 1.1 M ammonium sulfate) were looped and directly flash-frozen without cryoprotection.

### X-ray data collection and refinement

X-ray data sets were collected on the NSLS-II AMX 17-ID-2 beamline at Brookhaven National Laboratory (Upton, NY) at 100 K and 0.920100 Å wavelength. Initial data processing and reduction was completed automatically using XDS software version June 30, 2023[52]. Preprocessed data was further cut and scaled in CCP4i2 8.0.011[53] using Aimless. For structure determination, initial phasing was performed via molecular replacement with maximum-likelihood search procedures in the PHASER-MR module of the Phenix suite using a truncated 2-stack parallel RNA G-tetrad model derived from PDB structure 7SXP. Iterations of refinement and model building/editing were run with phenix.refine 1.20.1[54] and Coot 0.9.8.92[55] respectively.

### MD simulation

We used the initial x-ray structure coordinates of NS5-B M1 for the simulation. We conducted an MD simulation for the WT to compare the dynamics of M1 and WT. During the WT MD, we substituted two G→A changes (Fig. 1a and Table S1) in the initial x-ray structure. Simulations were run in explicit solvents where the G-quadruplex structures were placed in a truncated octahedron box filled with TIP3P water[56] and 0.15 M KCl using the force field OL3 for RNA in AMBER16 software[57]. The MD simulations were performed by using the Sander module with the SHAKE algorithm[58] applied to constrain all bonds involving hydrogen atoms with an integration time step of 2 fs. In the multistage equilibration protocol, the system was relaxed with 500 steps of steepest-descent energy minimization. The temperature of the system was increased from 0 K to 310 K for over 10 ps under constant-volume conditions. In the final step, the production runs on the system were subsequently performed for 1 μs under NPT (constant-pressure) conditions on the PMEMD CUDA module of AMBER16[57,59]. Trajectories were post-processed using the CPPTRAJ module of AMBERTOOLS16 to produce 50000 snapshots for analysis and visualization in UCSF Chimera 1.17 and ChimeraX 1.7 visualization software[60,61].

### Sequence alignment

NS5-B sequences were selected from the NCBI-deposited genomes of phylogenetically diverse members of the genus *Orthoflavivirus* (Table S4). Logos were generated using Weblogo 3.7.11[62].

### Reporting summary

Further information on research design is available in the Nature Portfolio Reporting Summary linked to this article.

## Data availability

The atomic coordinates and structure factor data generated in this study have been deposited in the Protein Data Bank (PDB) database under accession code 8UTG. The genome sequence data used to make Fig. 4. in this study are available in the NCBI database under the accession codes provided in Table S4. The structure of WNV NS5 methyltransferase domain used to make Fig. 4 is available in the PDB under accession code 2OY0. The MD coordinates and trajectory files are available at Figshare [https://doi.org/10.6084/m9.figshare.25854529]. The MD atomic distance, mass spectrometry, and CD data generated in this study are provided in the Source Data files. Source data are provided with this paper.

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

## Acknowledgements

This research used the AMX 17-ID-2 beamline of the National Synchrotron Light Source II, a U.S. Department of Energy (DOE) Office of Science User Facility operated for the DOE Office of Science by Brookhaven National Laboratory under Contract No. DE-SC0012704. The Center for BioMolecular Structure (CBMS) is primarily supported by the National Institutes of Health, National Institute of General Medical Sciences (NIGMS) through a Center Core P30 Grant (P30GM133893), and by the DOE Office of Biological and Environmental Research (KP1607011). Funding for this research was provided by awards from the National Institutes of Health, NIAID 1U19AI171403 (Project 4), M.A.B. and M.W.G., NIGMS GM111749, D.W, NIGMS GM137160 and NHLBI HL155178, G.M.K.P. and the National Science Foundation, MCB 2028902, G.M.K.P. and M.W.G. We thank Dr. Edwin Ogbonna for his contributions and aid in the procurement of crystallographic data from BNL. We thank Dr. Siming Wang and Dr. Wen Lu for their aid in mass spectrometry data collection.

## Author contributions

T.T.L., J.L.S., M.W.G., and W.D.W. conceived the project, selected the sequences studied and designed the experiments. T.T.L., M.W.G., J.L.S., and A.P. performed biophysical characterization. J.R.T. and T.T.L. performed X-ray crystallography sample preparations. J.R.T. performed X-ray crystallography, data analysis and model building. A.P. performed molecular dynamic simulations. J.L.S., A.P. T.T.L., and G.M.K.P. created the figures for the manuscript. T.T.L., A.P., J.L.S., and J.R.T. wrote the manuscript. M.A.B., M.W.G., G.M.K.P., and W.D.W. provided funding and edited the manuscript.

## Competing interests

The authors declare no competing interests.
