## [Peer Review File · Nature Communications]

REVIEWER COMMENTS

Reviewer #1 (Remarks to the Author):

In this manuscript titled “Structure of an RNA G-Quadruplex from the West Nile Virus Genome” the authors solve a crystal structure of a G-quadruplex containing sequence from the WNV RNA genome. This structure has significance as G-quadruplex binding compounds have been shown to have an antiviral effect and with a structure more specific compounds to WNV G-quadruplexes could be synthesized. These G-quadruplexes were shown to form in vitro and possess additional stabilizing capping structures that aid in the folding of the two-stack quadruplex. The stability and importance of these cap structures was validated through MD simulations and serve as a basis to design specific compounds to this viral G-quadruplex. The major significant contribution of this paper is that it highlights the potential for viral RNA based G-quadruplexes to have unique folds and offers a platform for the design of specific antiviral drugs. Overall, the data is compelling, exceptionally well presented graphically, and the conclusions are generally supported.

Minor Comments:

1. The introduction lacks sufficient discussion on the potential role of G-quadruplexes in the viral genome. Expanding on this aspect would enhance the paper's clarity and convey the importance of the structure to readers.
2. The reviewer suggests including a figure illustrating the entire G-quadruplex with highlighted regions of the capping structures, rather than relying solely on zoomed images.

Major Comment:

The major concern centers around the stability of the two-stack quadruplex observed in the crystal structure. The authors acknowledge that two-stack quadruplexes are less energetically stable than three or four-stack quadruplexes. The crystal packing forces a pseudo four-stack quadruplex, potentially providing false stability to the structure. While molecular dynamics (MD) simulations support the existence of the two-stack quadruplex, the reviewer recommends additional experimental evidence to further convince readers, particularly in the context of designing antiviral agents. The suggested experiment involves performing Circular Dichroism (CD) spectroscopy on G-quadruplex sequences with mutations in the capping structures, aiming to disrupt hydrogen bonds or pi-stacks stabilizing the structure. Additionally, analyzing these sequences through Nuclear

Magnetic Resonance (NMR) or Thioflavin T (ThT) assays under altered conditions could provide insights into the quadruplex's formation.

Reviewer #2 (Remarks to the Author):

The manuscript “Structure of an RNA G-Quadruplex from the West Nile Virus Genome” describes the NS5-B G-quadruplex crystal structure solved 1.97 Å. The authors have discussed the crystallography part in sufficient detail. They also performed NMR, CD, and mass-spectrometry experiments to support the relevance of their crystal structure in solution. However, the manuscript lacks a clear discussion about biological significance. Several structural studies on DNA and RNA G-quadruplexes have been published, and unlike claimed by the authors, some functional RNA G-quadruplexes, such as in Spinach RNA aptamer, have a unique fold with two G-tetrads. The results obtained with the isolated 21-nt sequence have been justified, but how this structure is relevant in the larger sequence context of the viral genome remains elusive. While the manuscript details an important viral RNA structure and given the scarcity of the viral RNA structural work in the literature, this manuscript has value in advancing our understanding of the RNA structural landscape; this study should be complemented with biologically significant results. Therefore, I am not entirely convinced that this manuscript, as is, is suitable for publication in Nature Communications.

1. While it is true that the authors report the first crystal structure of a G-quadruplex derived from a viral RNA genomic sequence. Nevertheless, the claim that such a structure with two G-tetrads is unique and novel is vague. Multiple RNA G-quadruplex structures with sophisticated folds with two G-tetrads, such as spinach RNA aptamer, have been reported previously. It would be more relevant to compare such two G-tread RNA structures and discuss the structural and functional diversities.

2. A discussion of crystal contacts, if any, should be presented. Are there any contacts between chains A and B in the crystal that might influence the observed dynamicity of the structure?

3. Authors should present the structures with a 2Fo-Fc or simulated omit map in the manuscript or supplementary information.

4. The crystal structure has two mutations, G2A and G20A. The authors discussed that the terminal nucleotides are essential for the overall stability of the NS5-B G-quadruplex structure. The manuscript lacks a comprehensive and transparent discussion about how these mutations affect the structure and, probably, the function of this RNA.

5. More importantly, the manuscript needs to clarify how the structure and function of this NS5-B G-quadruplex are modulated in the context of a larger viral genome sequence.

6. The PDB validation report lists NH₄⁺ as a “ligand of interest”. However, the relevance of this information is not provided in the manuscript or supplementary information. What is the basis for modeling the NH₄⁺ ion in the crystal structure?

Reviewer #3 (Remarks to the Author):

The manuscript focuses on potential G-quadruplex sites identified in the genomes of DNA and RNA viruses that have been proposed as regulatory elements for different stages of the viral life cycle. Biophysical studies have confirmed the ability of some G-rich sequences located in the genomes of Zika virus and tick-borne encephalitis virus to form tetrahelical G-quadruplexes. In vitro treatment of cells infected with Zika virus or tick-borne encephalitis virus with known G-quadruplex binders has shown antiviral activity. The authors present a high-resolution structure of the NS5-B quadruplex from the West Nile virus genome. This structure is the first crystal structure of a G-quadruplex derived from a viral RNA genome sequence. Structural analysis reveals the formation of two stacked G-tetrads that are further stabilized by a stacked base triad and transient non-canonical base pairing. The manuscript exhibits some degree of novelty as it expands the landscape of RNA quadruplex structures. It highlights the diversity and complexity of biological G-quadruplexes. The discovered structure could serve as a model for a conserved antiviral target in Orthoflavivirus genomes.

The results presented in the manuscript are timely. The presentation is clear, although not fully synchronized with the visual material and experimental results. The clarity of the graphics, such as in Figs. 1e and 2b, needs to be greatly improved. In the current version, many of the graphs show too many overlapping elements that are indescribable (see examples also below). To further improve the study, it is advisable to address certain challenges and consider changes as described below.

Authors are urged to acknowledge and cite crystal structures of RNA G-quadruplexes such as the one presented in *Nat. Commun.* recently, doi:10.1038/s41467-023-38683-3, and related RNA aptamers that form G-quadruplexes. What are the structural differences between RNA G-quadruplexes? Is the structure described here really specific for viral RNA genome sequence?

The 1D ¹H imino-proton spectrum of NS5-B M1 in the presence of K⁺ ions shows characteristic quadruplex signals at 10-12 ppm, that clearly do not correspond to a single structure. Statement on

'conformational homogeneity' needs to be corrected to be consistent with the experimental data. Why was the signal at δ 9.3 ppm omitted in Figure 1d, while it is shown in Figure S1a? Does the signal in the Watson-Crick spectral region at δ 13.1 ppm match the corresponding base pair from the X-ray crystal structure?

The sample temperature at which the spectrum shown in Fig. 1d was recorded must be indicated in the figure legend.

Do the NMR data and the complexity of the ^1H spectra shown in the manuscript correspond to the crystallographically determined dimeric structure in equilibrium with a monomeric species in solution? A simple concentration-dependent study could be very informative.

How can the two oligos be considered 'highly similar' if they differ in mobility on a PAGE gel (Fig. S1c) in both ways of visualization? Surprisingly, the WT oligo appears more homogeneous there.

Assuming that a number on the mass spectrometry printout in Figure 1b corresponds to a molecular ion (6830), it is not clear how it was rounded down to 'a monomer with a mass of ~6kDa', and not ~7kDa. This figure also needs to be corrected in the main text.

Add residue labels in Fig. 1e to add meaning and clarity with respect to the focus of the manuscript.

It would be helpful if the orientations of the overall structures in Figures 1e, 3c and 4c matched.

Although the overall structures may be similar, the claim that interactions involving G20 or A20 have little effect is speculative and clearly not supported by 'real' experimental data. It is simply unbelievable that A6-A20 and A6-G20 base pairs should have the same thermodynamic properties in terms of H-bonding and stacking as claimed by the authors.

The following sentence (A water network and interactions with the potassium ion further facilitate the interaction.) is simply confusing. The positions of the water molecules and the respective occupancies are neither presented nor discussed in the manuscript. How do K^+ ions facilitate the A6-A20 and A6-G20 interactions?

How can 'inter-asymmetric unit hydrogen bonding between A20 and A2' be expected when the former is H-bonded with A6? Anyhow, the presentation of inter-quadruplex interactions needs to be improved.

Formally speaking, there are only three (!) loops that connect the guanines involved in G-tetrads: A6-A9, C12 and U16. There is no justification to split the first four residue loop into two loops, although A6 and G8 are part of a base pair and triad, respectively.

Furthermore, in the G-quadruplex field, the term 'parallel loop' is not used. Term parallel is used solely to describe orientation of G-tracks. The authors should attempt to follow the accepted terminology in the field.

How can a stacked position of G8 be 'entropically' favorable when G8 is both stacked and H-bonded within a base triad?

It is not true that 'the short loop sequences contribute favorably to the overall energetics'. Rather, the reverse is true: the formation of base pairs and base triads that effectively stack on the nearby G-tetrads contributes to stability.

What data support the statement that 'a 4-purine stack G4-G5-G8-A6' is conserved in the 3D structure adopted by the wild type sequence?

Minor: With regard to the structure, A6 and A7 are at the 3' end and NOT at the 5' end, as indicated on p. 4 (cf. fig. 4c).

It is confusing that K⁺ ions and nucleobases are not shown at the same scale in a particular figure, such as in Fig. 3a left and right. Moreover, it is not even clear why the K⁺ ion is shown there, as it does not contribute to GUU or AA base pairing.

The clarity of the distance plots between the RNA bases at the triad in Fig. S3b needs to be improved. The blue and green lines cannot be resolved (from the red line).

We thank the reviewers for their thorough and insightful comments. We have taken advantage of the thoughtful suggestions and comments which have significantly strengthened the manuscript. In addition to changes in the text and Figures 1, 2, 3 and 4 of the manuscript, new data is presented in the supplementary material. Figures S2, S3, S4, S9, S10 and S11 are all new while S5, S6, S7 and S8 have been modified. Our responses are below in blue with excerpts of the manuscript quoted in italics with word changes highlighted in cyan.

Reviewer #1 (Remarks to the Author):

In this manuscript titled “Structure of an RNA G-Quadruplex from the West Nile Virus Genome” the authors solve a crystal structure of a G-quadruplex containing sequence from the WNV RNA genome. This structure has significance as G-quadruplex binding compounds have been shown to have an antiviral effect and with a structure more specific compounds to WNV G-quadruplexes could be synthesized. These G-quadruplexes were shown to form in vitro and possess additional stabilizing capping structures that aid in the folding of the two-stack quadruplex. The stability and importance of these cap structures was validated through MD simulations and serve as a basis to design specific compounds to this viral G-quadruplex. The major significant contribution of this paper is that it highlights the potential for viral RNA based G-quadruplexes to have unique folds and offers a platform for the design of specific antiviral drugs. Overall, the data is compelling, exceptionally well presented graphically, and the conclusions are generally supported.

Minor Comments:

1. The introduction lacks sufficient discussion on the potential role of G-quadruplexes in the viral genome. Expanding on this aspect would enhance the paper's clarity and convey the importance of the structure to readers.

We agreed and have modified the introduction.

2. The reviewer suggests including a figure illustrating the entire G-quadruplex with highlighted regions of the capping structures, rather than relying solely on zoomed images. *We agreed and have simplified Figure 1e to only portray a single copy of the structure from the crystallographic unit. We have annotated the tetrad and capping structures and have included labels for each nucleotide.*

Major Comment:

The major concern centers around the stability of the two-stack quadruplex observed in the crystal structure. The authors acknowledge that two-stack quadruplexes are less energetically stable than three or four-stack quadruplexes. The crystal packing forces a pseudo four-stack quadruplex, potentially providing false stability to the structure. While molecular dynamics (MD) simulations support the existence of the two-stack quadruplex, the reviewer recommends additional experimental evidence to further

convince readers, particularly in the context of designing antiviral agents. The suggested experiment involves performing Circular Dichroism (CD) spectroscopy on G-quadruplex sequences with mutations in the capping structures, aiming to disrupt hydrogen bonds or pi-stacks stabilizing the structure. Additionally, analyzing these sequences through Nuclear Magnetic Resonance (NMR) or Thioflavin T (ThT) assays under altered conditions could provide insights into the quadruplex's formation.

We thank the reviewer for raising an important point and suggesting relevant supporting experiments. Imino-proton spectra recorded for both WT and the M1 mutant at increasing temperatures showed the persistence of quadruplex specific resonance up to 328K (Figure S1a). In addition, we have designed a set of mutants (Table S3) with either a substitution or deletion in the dyad/triad region to assess their impact on structure and stability. We have characterized these mutants by CD spectra and CD melting, NMR spectroscopy, and gel electrophoresis (Figures S9-11) and have included interpretation of the results of these studies in the main body of the manuscript.

Reviewer #2 (Remarks to the Author):

The manuscript “Structure of an RNA G-Quadruplex from the West Nile Virus Genome” describes the NS5-B G-quadruplex crystal structure solved 1.97 Å. The authors have discussed the crystallography part in sufficient detail. They also performed NMR, CD, and mass-spectrometry experiments to support the relevance of their crystal structure in solution. However, the manuscript lacks a clear discussion about biological significance. Several structural studies on DNA and RNA G-quadruplexes have been published, and unlike claimed by the authors, some functional RNA G-quadruplexes, such as in Spinach RNA aptamer, have a unique fold with two G-tetrads. The results obtained with the isolated 21-nt sequence have been justified, but how this structure is relevant in the larger sequence context of the viral genome remains elusive. While the manuscript details an important viral RNA structure and given the scarcity of the viral RNA structural work in the literature, this manuscript has value in advancing our understanding of the RNA structural landscape; this study should be complemented with biologically significant results. Therefore, I am not entirely convinced that this manuscript, as is, is suitable for publication in Nature Communications.

1. While it is true that the authors report the first crystal structure of a G-quadruplex derived from a viral RNA genomic sequence. Nevertheless, the claim that such a structure with two G-tetrads is unique and novel is vague. Multiple RNA G-quadruplex structures with sophisticated folds with two G-tetrads, such as spinach RNA aptamer, have been reported previously. It would be more relevant to compare such two G-tread RNA structures and discuss the structural and functional diversities.

We thank the reviewer for raising this point and agree that a direct comparison of this structure to a previously reported two-stack RNA G-quadruplex improves the clarity of this paper. We have cited Banco, M. T., & Ferré-D'Amaré, A. R. (2021). The emerging structural complexity of G-quadruplex RNAs. *RNA*, 27(4), 390-402. in the initial version of the manuscript, but we agree that a more comprehensive discussion is warranted

and that our previous version did not clearly highlight these important structures. We have included a comparison:

“Some examples of two-stack RNA tetrads with additional stabilizing features are Mango-III³⁴ and Spinach^{35, 36} RNA aptamers. In the Mango-III aptamer, two G-tetrads stack on a noncanonical terminal U-A-U base triple and are further capped by a U-A base pair at the top of the tetrads³⁴. Similarly, the Spinach aptamer has a two-stack G-tetrad^{35, 36}, which stacks on a noncanonical G-U-U-C tetrad³⁶ and is further capped by a U-A-U base triple. Although the monomeric NS5-B M1 is structurally distinct from the vegetable family of aptamers that form in duplex RNA, its stability is also markedly enhanced by unusual terminal structures.” (pg. 3)

2. A discussion of crystal contacts, if any, should be presented. Are there any contacts between chains A and B in the crystal that might influence the observed dynamicity of the structure?

We have included a figure (Figure S3) to illustrate the crystal contacts. We have also included a supplemental discussion of the contacts in the context of dynamics within the crystal structure. (pg. 22-23)

3. Authors should present the structures with a 2Fo-Fc or simulated omit map in the manuscript or supplementary information.

We have included this figure which also clarifies the cations (Figure S4) as well as the data file. (pg. 24)

4. The crystal structure has two mutations, G2A and G20A. The authors discussed that the terminal nucleotides are essential for the overall stability of the NS5-B G-quadruplex structure. The manuscript lacks a comprehensive and transparent discussion about how these mutations affect the structure and, probably, the function of this RNA.

We agree that the capping structures are important to the overall stability of the quadruplex and that the mutations present in the M1 structure may cause differences in the thermodynamics. However, the similar NMR spectra and persistence of proton signals at 328K (Figure S1a) for both sequences indicate that both structures have similar stability. We agree with the reviewer that further exploration of these differences is warranted and have therefore recorded CD melting experiments with the WT and M1 sequences as well as a set of mutants (Table S3). We have observed that both the WT and M1 have very similar thermal melting temperatures (T_m $47 \pm 1^\circ\text{C}$ vs T_m $48 \pm 1^\circ\text{C}$, Figure S11). Furthermore, the CD thermal stability experiments show that deletions in the dyad (M1 D2) or triad (M1 D3) and especially the triad mutant (M1 G8A) reduce the stability of the G-quadruplex structures (Figure S11). This indicates that the triad is important for the stability of this quadruplex structure. We also conclude that the G2A and G20A mutations have very little if any effect on the quadruplex stability and that M1 and WT are indeed similar as evidenced by their T_m .

5. More importantly, the manuscript needs to clarify how the structure and function of

this NS5-B G-quadruplex are modulated in the context of a larger viral genome sequence.

We wholeheartedly agree that understanding the structure and function of NS5-B in the context of the viral genome sequence is an important next step. Unfortunately, much longer sequences are less likely to be amenable for either crystallographic or NMR studies and would not reflect their impact in the context of the full genomic RNA. We are however continuing to work to characterize WNV quadruplexes in the full ~11kb genome. We hope to share this work in a future publication.

6. The PDB validation report lists NH₄⁺ as a “ligand of interest”. However, the relevance of this information is not provided in the manuscript or supplementary information. What is the basis for modeling the NH₄⁺ ion in the crystal structure?

We thank the reviewer for the previous suggestion to include the 2Fo-Fc map (#3, Figure S4) as it provides important context for the inclusion of NH₄⁺ in the crystal structure. The ions at the top and bottom of the quadruplex are likely to exchange. Given that the quadruplex was annealed with 50 mM KCl and then crystallized in 1 M NH₄⁺, it is reasonable that some the ions have exchanged. The alternating sizes of densities and the reduction in R_{free} with the displayed configuration informed the fitting of the ions.

Reviewer #3 (Remarks to the Author):

The manuscript focuses on potential G-quadruplex sites identified in the genomes of DNA and RNA viruses that have been proposed as regulatory elements for different stages of the viral life cycle. Biophysical studies have confirmed the ability of some G-rich sequences located in the genomes of Zika virus and tick-borne encephalitis virus to form tetrahelical G-quadruplexes. In vitro treatment of cells infected with Zika virus or tick-borne encephalitis virus with known G-quadruplex binders has shown antiviral activity. The authors present a high-resolution structure of the NS5-B quadruplex from the West Nile virus genome. This structure is the first crystal structure of a G-quadruplex derived from a viral RNA genome sequence. Structural analysis reveals the formation of two stacked G-tetrads that are further stabilized by a stacked base triad and transient non-canonical base pairing. The manuscript exhibits some degree of novelty as it expands the landscape of RNA quadruplex structures. It highlights the diversity and complexity of biological G-quadruplexes. The discovered structure could serve as a model for a conserved antiviral target in Orthoflavivirus genomes.

The results presented in the manuscript are timely. The presentation is clear, although not fully synchronized with the visual material and experimental results. The clarity of the graphics, such as in Figs. 1e and 2b, needs to be greatly improved. In the current version, many of the graphs show too many overlapping elements that are indescribable (see examples also below). To further improve the study, it is advisable to address certain challenges and consider changes as described below.

We thank the reviewer for their thorough and thoughtful review of our manuscript and have sought to improve the overall clarity of our figures by implementing their suggestions. (See below)

Authors are urged to acknowledge and cite crystal structures of RNA G-quadruplexes such as the one presented in Nat. Commun. recently, doi:10.1038/s41467-023-38683-3, and related RNA aptamers that form G-quadruplexes. What are the structural differences between RNA G-quadruplexes? Is the structure described here really specific for viral RNA genome sequence?

We thank the reviewer for raising this point. Although we previously cited Banco, M. T., & Ferré-D'Amaré, A. R. (2021). The emerging structural complexity of G-quadruplex RNAs. RNA, 27(4), 390-402 in the initial version of the manuscript, we agree that a more comprehensive discussion is warranted and regret that our previous version did not clearly highlight these important structures. We have included a description and citations for the Mango-III and Spinach aptamers which are two-stack quadruplexes but have chosen not to highlight the Beetroot structure as it is a homodimer of four-stack quadruplexes composed of three G-tetrads and a noncanonical G-U-A-A tetrad. Please see pg. 3 for a description of these structures.

The 1D ¹H imino-proton spectrum of NS5-B M1 in the presence of K⁺ ions shows characteristic quadruplex signals at 10-12 ppm, that clearly do not correspond to a single structure. Statement on 'conformational homogeneity' needs to be corrected to be consistent with the experimental data.

We agree and did not mean to imply that there was a single structure present. Therefore, the term 'conformational homogeneity' was misleading and have corrected the manuscript to reflect this. In the revised manuscript we have rewritten the sentence as following,

"NS5-B M1 exhibited the same stability and a similar but better-resolved imino proton spectrum, indicating less conformational variability of the monomeric structures (Figure 1d, S1a)." (pg. 2)

Why was the signal at d 9.3 ppm omitted in Figure 1d, while it is shown in Figure S1a? Does the signal in the Watson-Crick spectral region at d 13.1 ppm match the corresponding base pair from the X-ray crystal structure?

Figure 1d shows a spectral range corresponding to 10-13.7 ppm. This is a figure that contains 5 different panels and space is therefore limited. We had included a wider spectral window (9-15ppm) in Figure S1a to demonstrate thermal stability and a direct comparison to the WT. Both M1(G2A, G20A) and WT show a signal at 13.1ppm. It is tempting to attribute this signal to the base pair in the triad. To help determine the source of this signal we performed NMR on two deletion mutants, D2 and D3, and one triad substitution mutant G8A (Figure S10) and have included relevant discussion: *"NMR imino proton spectra of the mutants exhibited characteristic G-tetrad signals at 10-12 ppm (Figure S10). Noticeably, while the M1 D2 spectrum retains a signal around 12.8 ppm, there were no signals observed in the Watson-Crick base pairing region of the spectra for either M1 D3 or M1 G8A. These data suggest that the imino proton*

signals in the 12-14 ppm region likely arise from the G·U base pair in the g-triad.” (pg. 4)

The sample temperature at which the spectrum shown in Fig. 1d was recorded must be indicated in the figure legend.

We appreciate the reviewer's suggestions and in the revised manuscript we have added the experimental temperature.

“d) The 1D ¹H imino proton spectrum of NS5-B M1 at 298K in the presence of K⁺ ions exhibits characteristic quadruplex signals at 10-12 ppm and is consistent with the spectrum of the WT sequence (Figure S1a).” pg. 7

Do the NMR data and the complexity of the ¹H spectra shown in the manuscript correspond to the crystallographically determined dimeric structure in equilibrium with a monomeric species in solution? A simple concentration-dependent study could be very informative.

We apologize that our original portrayal in Figure 1e could be interpreted as a dimer. While two copies of the quadruplex do pack into a single asymmetric unit, there are not sufficient contacts between copies for the quadruplex to be interpreted as a dimer. We've adjusted our depiction in this revised manuscript in 1e to only show a single copy but have provided a full map of the asymmetric unit with mapped crystal contacts (Figure S3). pg. 22

We also thank the reviewer for the helpful suggestion of including a concentration-dependent NMR study. We have conducted an extended concentration-dependent study of the imino protons in H₂O as well as aromatic protons in D₂O of the WT and M1 quadruplex (Figure S2). Both studies demonstrate that the spectra are not concentration dependent consistent with our finding that the quadruplex is a monomer in solution. pg. 21

How can the two oligos be considered 'highly similar' if they differ in mobility on a PAGE gel (Fig. S1c) in both ways of visualization? Surprisingly, the WT oligo appears more homogeneous there.

All the biophysical data (NMR, CD, CD-Tm) indicate that the two sequences are similar. While there is a small mobility difference on native PAGE, the difference in mobility is consistent with small changes in the loop and the resulting loop flexibility. The M1 G8A mutant which perturbs the triad demonstrates a much larger difference in mobility on native PAGE despite only differing from the M1 sequence by one nucleotide. This further supports that the A2G and A20G mutations present in M1 have a smaller effect on the overall structure. We have modified the description of the PAGE mobility in the manuscript:

“Both wild-type and M1 NS5-B fold into compact structures that exhibit similar electrophoretic mobility on native PAGE and stain equally with the quadruplex-specific dye, Thioflavin T (ThT) (Figure S1c).” pg. 2

Assuming that a number on the mass spectrometry printout in Figure 1b corresponds to a molecular ion (6830), it is not clear how it was rounded down to 'a monomer with a mass of ~6kDa', and not ~7kDa. This figure also needs to be corrected in the main text.

We thank the reviewer for noticing this. It has been corrected in the main body of the manuscript and figure legends.

“Mass spectrometry (Figure 1c) confirmed the presence of a single ~7 kDa species....”
pg. 2

Add residue labels in Fig. 1e to add meaning and clarity with respect to the focus of the manuscript.

We have made the suggested changes to our figure.

It would be helpful if the orientations of the overall structures in Figures 1e, 3c and 4c matched.

We appreciate the suggestions and followed them by adding the residue labels and matching the structure orientations has increased the clarity of the manuscript.

Although the overall structures may be similar, the claim that interactions involving G20 or A20 have little effect is speculative and clearly not supported by ‘real’ experimental data. It is simply unbelievable that A6-A20 and A6-G20 base pairs should have the same thermodynamic properties in terms of H-bonding and stacking as claimed by the authors.

We agree that the mutations present in the M1 structure could cause slight differences in the thermodynamics of the structure due to differences in the dyad. However, the similar NMR spectra and persistence of proton signals at 328 K for both sequences (Figure S1a) indicate that both structures have similar stability. We agreed with the reviewer that further exploration of these differences was warranted and have completed CD melting experiments with the WT and M1 sequences as well as a set of mutants (Table S3). We have observed that both the WT and M1 have similar thermal melting points (T_m $47 \pm 1^\circ\text{C}$ vs T_m $48 \pm 1^\circ\text{C}$, Figure S11). We therefore conclude that the G2A and G20A mutations have minimal contributions to the quadruplex stability and that M1 and WT are indeed similar while not identical.

The following sentence (A water network and interactions with the potassium ion further facilitate the interaction.) is simply confusing. The positions of the water molecules and the respective occupancies are neither presented nor discussed in the manuscript. How do K^+ ions facilitate the A6-A20 and A6-G20 interactions?

We agree with the reviewer that this sentence is confusing and have removed the sentence from the manuscript.

How can ‘inter-asymmetric unit hydrogen bonding between A20 and A2’ be expected when the former is H-bonded with A6? Anyhow, the presentation of inter-quadruplex interactions needs to be improved.

We thank the reviewer for bringing up this point and have included a full map of the crystal contacts in Figure S3. pg. 22

Formally speaking, there are only three (!) loops that connect the guanines involved in G-tetrads: A6-A9, C12, and U16. There is no justification to split the first four residue loops into two loops, although A6 and G8 are part of a base pair and triad, respectively.

We have adjusted our description of the loops to reflect this suggestion. We have removed the word “loop” from the labels in Figure 3c and adjusted our description of the A6-A9 and U16 loops with the following:

“The C12 loop consists of a single, one nucleotide segment (Figure 3c). The A6-A7-G8-A9 loop and the U15-U16 loop contain single nucleotide segments (U15, A7 and A9) between the capping structure residues.” pg. 4

Furthermore, in the G-quadruplex field, the term ‘parallel loop’ is not used. Term parallel is used solely to describe orientation of G-tracts. The authors should attempt to follow the accepted terminology in the field. How can a stacked position of G8 be ‘entropically’ favorable when G8 is both stacked and H-bonded within a base triad?

Thank you for catching this error and informing us of the correct terminology.

“To maintain the parallel topology in the NS5-B M1 structure, the A6 and A7 bases located at the top of the quadruplex create a striking structural feature. While A6 is involved in the terminal dyad capping structure, A7 makes a short turn to allow G8 to occupy a more favorable stacked position in the g-triad.” pg. 4

It is not true that ‘the short loop sequences contribute favorably to the overall energetics’. Rather, the reverse is true: the formation of base pairs and base triads that effectively stack on the nearby G-tetrads contributes to stability.

We did not mean to imply that a loop residue in the absence of any stacking or hydrogen bonding is a major contributor to stability. However, loop sequences contribute indirectly to stability in the sense that they allow or inhibit the optimal placement of the nucleotides in the triad and tetrad segments. We have modified the original manuscript to improve clarity:

“We hypothesize that the short nucleotide segments contribute favorably to the overall energetics³¹ and the crystal packing of this structure by constraining the placement of nucleotides in the triad and tetrad regions thereby facilitating optimal base stacking.” pg. 4

What data support the statement that ‘a 4-purine stack G4-G5-G8-A6’ is conserved in the 3D structure adopted by the wild type sequence?

Both M1 and WT contain identical residues for the 4-purine stack. Our MD simulations for both WT and M1 quadruplexes show that this 4-purine stack persists throughout the simulations. In addition, NMR and CD-T_m data show essentially identical stability for WT and M1 quadruplexes. We have modified our description of this feature as follows:

“This generates a 4-purine stack G4-G5-G8-A6 (Figure 3b) which we hypothesize also forms in the WT structure given the similar melting temperatures of the M1 and WT quadruplexes and its persistence in an MD simulation of the WT sequence.” pg. 4

Minor: With regard to the structure, A6 and A7 are at the 3’ end and NOT at the 5’ end, as indicated on p. 4 (cf. fig. 4c).

We agree that using the 3’ and 5’ terminology to describe the location of A6 and A7 may be confusing and have modified our description accordingly:

“To maintain the parallel topology in the NS5-B M1 structure, the A6 and A7 bases located at the top of the quadruplex create a striking structural feature.” pg. 4

It is confusing that K⁺ ions and nucleobases are not shown at the same scale in a particular figure, such as in Fig. 3a left and right. Moreover, it is not even clear why the K⁺ ion is shown there, as it does not contribute to GUU or AA base pairing.

We have adjusted the scaling between Figure 2 and Figure 3 to be consistent and have only included the ions relevant to each structure. For the tetrads, we have included the K⁺ ion. For the triad, we have included NH₄⁺. The inclusion of the ions is based on the distances in both the crystal structure and MD simulation. No ion was included for the dyad as distances were not sufficient to support contacts.

The clarity of the distance plots between the RNA bases at the triad in Fig. S3b needs to be improved. The blue and green lines cannot be resolved (from the red line).

We agree and have replotted these figures for improved clarity (Figures S5-S8).

REVIEWERS' COMMENTS

Reviewer #1 (Remarks to the Author):

The reviewer would like to thank the authors for their careful consideration of the comments provided and agree that the manuscript has been significantly strengthened. Overall, the additions made to the introduction and edits made to the figures throughout have greatly increased the overall clarity of the paper and better convey the significance of the study.

The reviewer would also like to thank the authors for performing additional experiments to analyze the two-stack quadruplex and is satisfied with the evidence presented. At this time, the reviewer has no additional comments for the authors, and we again thank them for their diligent efforts and thoughtful response.

Reviewer #2 (Remarks to the Author):

The authors have satisfactorily addressed most of my comments. However, the information about WNV quadruplex formation in longer RNA sequences is still lacking. I understand that X-ray crystallography or NMR studies with longer and potentially floppy structures would be pretty challenging; authors could have performed some CD studies or chemical footprinting assays (such as DMS footprinting) to show G-quadruplex formation in the longer RNA sequences.

Otherwise, the manuscript is now well-improved and suitable for publication in nature communications.

Reviewer #3 (Remarks to the Author):

The manuscript was improved by taking into account the suggestions of the reviewers. However, some of them were simply ignored without justification.

It is not clear why the signal at δ 9.3 ppm was omitted in Figure 1d, while it is shown in Figure S1a. The authors' response that Figure 1 contains 5 panels is irrelevant. It appears that the signal at δ 9.3 ppm has not been assigned to a specific H-bond. What if it is an important element of RNA structure and its stabilization in solution that has been overlooked so far?

p. 2; Rephrase 'four guanines Hoogsteen bond to form a tetrad' to clarify that four guanines form hydrogen bonds in Hoogsteen geometry to form a tetrad.

The authors are again encouraged to improve the clarity and reduce the overlap of the graphics, such as in Figure 1e. Only by reducing the overlap of structural elements, which are indescribable in the current version, can a reader understand this interesting structure.

What are the structural differences between RNA G-quadruplexes? Is the structure described here really specific to the viral RNA genome sequence?

We thank the reviewers again for their time and consideration of our manuscript.

REVIEWERS' COMMENTS

Reviewer #1 (Remarks to the Author):

The reviewer would like to thank the authors for their careful consideration of the comments provided and agree that the manuscript has been significantly strengthened. Overall, the additions made to the introduction and edits made to the figures throughout have greatly increased the overall clarity of the paper and better convey the significance of the study.

The reviewer would also like to thank the authors for performing additional experiments to analyze the two-stack quadruplex and is satisfied with the evidence presented. At this time, the reviewer has no additional comments for the authors, and we again thank them for their diligent efforts and thoughtful response.

Thank you!

Reviewer #2 (Remarks to the Author):

The authors have satisfactorily addressed most of my comments. However, the information about WNV quadruplex formation in longer RNA sequences is still lacking. I understand that X-ray crystallography or NMR studies with longer and potentially floppy structures would be pretty challenging; authors could have performed some CD studies or chemical footprinting assays (such as DMS footprinting) to show G-quadruplex formation in the longer RNA sequences.

Regarding longer sequences, we have confirmed the formation of the quadruplex in a 27-mer and a 30-mer sequence using a ThT gel (Figure S1a). We have also included some modifications to the manuscript:

We experimentally validated the formation of the predicted WNV RNA quadruplex NS5-B, first with a 30- and then 27-mer sequence, using gel electrophoresis and Thioflavin T (ThT) staining and confirmed monomer formation using mass spectrometry (Figure S1a). (pg. 2)

and

Experimental methods for probing quadruplexes in the context of longer biological sequences are limited by imaging capabilities and the specificity of quadruplex-specific antibodies and fluorescent stains and do not provide structural details. (pg. 6)

Otherwise, the manuscript is now well-improved and suitable for publication in nature communications.

Thank you!

Reviewer #3 (Remarks to the Author):

The manuscript was improved by taking into account the suggestions of the reviewers. However, some of them were simply ignored without justification.

It is not clear why the signal at δ 9.3 ppm was omitted in Figure 1d, while it is shown in Figure S1a. The authors' response that Figure 1 contains 5 panels is irrelevant. It appears that the signal at δ 9.3 ppm has not been assigned to a specific H-bond. What if it is an important element of RNA structure and its stabilization in solution that has been overlooked so far? The identity of the NMR peak at 9.3 ppm is difficult to determine without selective isotope labeling of many different RNA residues. In addition, both NS5B M1 and WT have multiple conformations of the monomer which greatly hampers detailed NMR work. This is evident from the number and different intensities of the imino proton resonances in the G quadruplex region which is one of the reasons why we pursued X-ray crystallography as an alternative method for determining structural information. We have addressed that in the manuscript.

Eight proton signals arising from the imino protons of two stacked G tetrads are expected in this region. However, more peaks of different intensity were observed indicating the presence of multiple conformations.

and

However, neither sequence yielded an NMR spectrum that was resolved enough for structure assignment, but NS5-B M1 formed suitable crystals that enabled us to solve the x-ray crystal structure of a WNV genomic RNA quadruplex (Figure 1e and Table 1).

p. 2; Rephrase 'four guanines Hoogsteen bond to form a tetrad' to clarify that four guanines form hydrogen bonds in Hoogsteen geometry to form a tetrad.

We have rewritten the sentence as:

G-quadruplexes form in G-rich DNA or RNA when four guanines form hydrogen bonds to create Hoogsteen geometry and form a tetrad.

The authors are again encouraged to improve the clarity and reduce the overlap of the graphics, such as in Figure 1e. Only by reducing the overlap of structural elements, which are indescribable in the current version, can a reader understand this interesting structure.

We have updated Figure 1e to reduce the overlap. In addition, we also produced a movie of the quadruplex to minimize overlap.

What are the structural differences between RNA G-quadruplexes? Is the structure described here really specific to the viral RNA genome sequence?

Given the limited number of RNA quadruplex structures currently, it is impossible to say how specific this structure is to the viral RNA genome sequence. We recognize that other RNA

quadruplex structures (many of them synthetic) have shown the presence of unusual capping structures and stabilizing hydrogen bonds with loop residues. At the time of writing, we were unable to find an RNA quadruplex structure that had a component identical to the features present in our structure. However, it is entirely possible that other structures with these features will be identified as both computational algorithms and as more RNA structures are determined.